# A QoS Based Adaptive Backoff Scheme for Vehicular Ad Hoc Networks

**DOI:** 10.3390/s18124421

**Published:** 2018-12-14

**Authors:** Tianjiao Zhang, Qi Zhu

**Affiliations:** 1Key Lab on Wideband Wireless Communications and Sensor Network Technology of Ministry, Nanjing University of Posts and Telecommunications, Nanjing 210003, China; 2014010204@njupt.edu.cn; 2Jiangsu Key Lab of Wireless Communications, Nanjing University of Posts and Telecommunications, Nanjing 210003, China

**Keywords:** VANETs, QoS, backoff scheme, priority-based

## Abstract

The media access control (MAC) protocol is a key element in the design of vehicular ad hoc networks (VANETs) that directly affects the network performance. The backoff schemes of existing MAC protocols apply the single backoff process and therefore are not suitable for multi-class data transmission. Additionally, they cannot satisfy the delay requirements of emergency data in the case of varying number of vehicles, causing an adverse effect to the intelligent transportation system (ITS). This paper presents a priority-based adaptive backoff scheme that can enhance the binary exponential backoff (BEB) algorithm as well as the polynomial backoff (QB) algorithm. This system distinguishes priority data with different delay requirements first and designs different backoff schemes for each type of data later. The two-dimensional Markov Chain is used to analyze the backoff scheme and determine the expressions for throughput and delay. The simulation results show that the backoff scheme provided by this paper can reduce the average data delay and regulate each kind of data delay adaptively, according to the varying number of vehicles and different delay requirements.

## 1. Introduction

VANET is a special mobile network that applies the philosophy of traditional mobile ad hoc networks (MANETs) to road traffic [1]. VANETs have different characteristics and problems compared to MANETs, such as the actual road scene, varying number of nodes, and high mobility of nodes. Further, VANETs are less tolerant to delays than MANETs because a delay affects driving safety directly. Therefore, a MAC protocol with a guaranteed delay for VANETs is worth researching.

IEEE 802.11 is the standard of wireless networks currently [2]. It consists of a series of protocols, such as 802.11a and 802.11b. Among these, IEEE 802.11p is defined for VANETs [3]. It enables data transmission between moving vehicles and roadside units (RSUs). It uses the enhanced distributed channel access (EDCA) mechanism, similar to IEEE 802.11e. The QoS guarantee comes with a host of challenges [4]. The network nodes access the channel using the carrier sense multiple access with collision avoidance CSMA/CA mechanism, and execute the BEB, when a channel collision occurs [5].

The BEB algorithm has several drawbacks, such as slow convergence speed, unfair channel contention, and the lack of a QoS guarantee for emergency data. Researchers have attempted to overcome these shortcomings in different ways, including using a two- or three-dimensional Markov Chain [6,7], employing a queuing network model, an ON/OFF model [8], and many other mathematical theories to analyze the algorithm performance and develop adaptive backoff algorithms. The service differentiation dynamic backoff (SDDB) algorithm [9] classifies data based on priorities. The contention window multiplicative increases, when a channel collision happens. It is reset to the minimum value, when the data with the highest priority is transmitted successfully. On the other hand, it will decrease linearly upon successful transmission of the data with low priority. However, the algorithm cannot be extended to networks with more than two kinds of data. In [10], the author introduces a distributed discrete congestion control algorithm that allows nodes to monitor the channel traffic load and adjust the transmission parameter dynamically. Thus, channel utilization and fairness of the nodes accessing the channel can be improved. The QB [11] and the BEB algorithms can both achieve the same maximum throughput; however, the second moment of access delay when using the QB algorithm is shorter than that when using BEB, allowing the former to achieve better queuing and fairness performance than the latter. Reference [12] analyzes the system performance with a finite retry limit and discovers that there is a correlation between network performance and retry time. The system delay will grow rapidly with a rise in retry time. If the retry time is set too small, the system-saturated throughput will go down. Therefore, it is important to set the retry time according to the system capacity. The performance of multi-rate CSMA networks is analyzed in [13]. When the network throughput and the sum rate are ensured, the method for initialization of the minimum contention window according to the number of nodes with different transmission rates is investigated. These studies [6,7,8,9,10,11,12], however, do not consider the network requirement with multiple kinds of data. Although [13] shows how to set the contention window with multiple types of data, there is no optimization of the backoff process. The FMC-MAC protocol [14] allows safety messages broadcasted on service channel and non-safety data transmitted on control channel in a flexible way. It not only guarantees the reliability of safety applications, but also improves the throughput of non-safety services. It cannot increase the throughput when the node density is high and it does not analyze the system delay, which is one of the important system performances in the vehicular network.

To satisfy the QoS requirements of all kinds of data in the vehicular networks, this paper proposes an adaptive backoff algorithm, based on traffic priority. This algorithm can decrease the average data delay and ensure that all types of data meet their own transmission requirements. In the system, data are categorized as per their delay requirements. The data with higher delay requirement, for example, emergency data driving information, have higher transmission priority. The data with lower delay requirement, like user wanting to download the video services, has lower transmission priority. Each type of data has its own delay limit, according to which the transition parameters are optimized. Moreover, each type of data has its own backoff scheme, and the system can optimize the delays of any kinds of data adaptively. The simulation results show that this algorithm can optimize the average total delay, thereby controlling the throughput and delay of different kinds of data.

The innovative aspects of this algorithm are as follows: (1) We present an adaptive transmission algorithm for vehicular ad hoc networks, which can incorporate multi-type data transmission. By using the adaptive backoff scheme, we can obtain two new backoff models, called the enhanced BEB algorithm (eBEB) and the enhanced QB algorithm (eQB). The transmission parameters are optimized according to the delay requirements and the number of vehicles in order to achieve the best network performance; (2) The transmission probability of the backoff process is controllable in real-time. This parameter is calculated using the recorded conflicting information so that an emergency situation within the system does not cause the contention window to shock; (3) A two-dimensional Markov Chain is established to analyze the performance of the backoff algorithm. Expressions for network throughput and delay are derived, and the correctness of the derivation is verified through a simulation; (4) It is proposed that time delay is caused due to an optimization problem and particle swarm algorithm has been used to resolve the same. The transition parameters are optimized to achieve the delay requirements.

The remainder of this paper is organized into sections. Section 2 establishes the system model that includes suggesting how to estimate the number of contention nodes and determine the process of data transmission. The model of backoff scheme is presented in Section 3. It also comprises information on the backoff processes of different kinds of data and calculating the backoff transition probability. The two-dimensional Markov Chain is shown in Section 4 in order to derive the expressions for network throughput and delay. Section 5 provides the simulation results to verify the feasibility of the algorithm. Finally, concluding remarks are summarized in Section 6.

## 2. System Model

### 2.1. Estimating the Number of Contention Nodes

One of the most important characteristics of vehicular networks is the real-time variability of the number of nodes in the system, which greatly affects network transmission performance. Assume that the distribution of the vehicles in the vehicular network is Poisson Point Process (PPP), B(λ,r). For any given vehicle, the number of neighbor nodes follows $P\left(n=k\right)=\frac{\lambda^{k}e^{-\lambda }}{k!},\;k=0,1,2,\cdots$. The vehicles run on a two-way, four-lane highway, in order to facilitate the calculation. Ignoring the influence of the width of the roads on the transmission coverage, we can know the excepted number of neighbor nodes is λ=4·2r·α. In the formula, α is the distribution density and *r* is the communication range.

System parameters, such as the maximum backoff stage *m*, the minimum contention window $W_{0}$, the transmission speed, etc., are provided to a particular network and cannot be changed. A change in the number of nodes in the network triggers a change in the traffic load. While these parameters cannot change according to the traffic load, the network performance will go down, with the change in the number of vehicles.

The channels in the vehicular network are divided into a control channel and a service channel. Each vehicle node has a unique MAC address, called $MAC_{ID}$, which can be used to identify the vehicle. Each node broadcasts a beacon message periodically on the control channel to other nodes within its transmission range, including the nodes in the beacon message. Other vehicle nodes can identify this certain node after receiving the beacon message. Every node monitors the control channel. After receiving the beacon messages from other nodes, they pick up the respective s and include them into their one-hop neighbor list. The time received is recorded into the list at the same time. Each node maintains the one-hop neighbor list by monitoring the channel. The various nodes in the network broadcast the beacon messages periodically and join the other nodes’ one-hop neighbor list repeatedly. When a certain node finds that other nodes are joining repeatedly, it records the new time received. If a certain node finds that access times of nodes in its one-hop neighbor list are too long, it will delete them from its list to prevent the list from increasing infinitely, as this condition implies that these nodes have moved from its transmission range.

The service channel is used by nodes for contention access and message transmission. As the data transmission time is too little for vehicles to move a very long distance, we can assume that all the vehicles can be seen still when they transmit or receive data. For example, the speed of the vehicle is 100 km/h, the data transmission time is level, the vehicle will move far less than 1 m. When a vehicle begins transiting data, it judges the distance and chooses a nearer relay node or not according to its one-hop neighbor list. So the transmission collision happens only if the channel is used by other nodes at the same time. To an individual node, all the nodes existing in its one-hop neighbor list can be seen as contention nodes. Although the system parameters cannot be changed, the vehicle nodes can control their own backoff processes. By estimating the number of contention nodes, the vehicle nodes can reasonably choose contention windows that will reduce the collision probability. This, in turn, can improve the network performance. Considering that congestion caused by vehicle flow typically remains for only a short duration and is paroxysmal, when estimating the number of contention nodes, we consider not only the number of nodes in the one-hop neighbor list, but also the previous number of contention nodes. Assuming that at a certain time *t*, the number of nodes in the one-hop neighbor list is $n_{n}\left(t\right)$ and the estimated number of contention nodes is n(t), then $n\left(t\right)=\frac{\sum_{i=1}^{t-1}n\left(i\right)+n_{n}\left(t\right)}{t}$.

### 2.2. Transmission Process

When new data arrives for transmission, the node performs various backoff processes according to the type of data, and chooses the backoff window randomly. Next, the node monitors the channel until the backoff process is complete and sends the data. If the data is transmitted successfully, the backoff stage of this kind of data is reset to zero with a certain probability. Otherwise, the backoff stage increases, the node chooses the backoff window again and waits to send the data. The transmission process is shown in Figure 1.

## 3. Backoff Algorithm

In VANET, the vehicle nodes communicate with other vehicles or roadside units and therefore, many types of data exist in the network. Different types of data have their own transmission delay requirements. For instance, emergency data, generated when vehicles crash, has a low delay tolerance, and must be sent immediately. The data from certain kinds of vehicles also have different priorities; special vehicles like police cars and fire engines have a higher priority to send data, compared to private cars. It is necessary to reduce the delay in high priority data by arranging all the data reasonably, according to their delay requirements.

Just like in the BEB algorithm, the nodes have m+1 backoff stages (0,1,⋯,m), $W_{0},W_{1},\cdots ,W_{m}$ expresses the contention window length of a certain backoff stage *i*. In BEB, $W_{i}=2^{i}W_{0}$ (where $W_{0}$ is the initialized window), and $CW_{\min}=2^{0}W_{0}$, $CW_{\max}=2^{m}W_{0}$ can express the minimum and maximum contention window sizes. In QB [11], $W_{i}={\left(1+i\right)}^{2}W_{0}$, $CW_{\min}=W_{0}$ and $CW_{\max}={\left(1+m\right)}^{2}W_{0}$. In IEEE 802.11e, EDCA algorithm defines eight kinds of traffic categories (TC) and four kinds of access categories (AC), which is also cited in DSRC/WAVE standard and IEEE 1609.3. Eight kinds of TC are mapped to the queues of four kinds of AC, respectively. Similarly to the definition of EDCA, in the assuming vehicular adhoc network, the data are divided into *K* types according to their delay requirements. For example, if the data are just divided into high priority and low priority, in this case, the *K* equals to two. If we need four ACs like EDCA, the *K* equals to four etc. The larger *K* is, the higher the data discrimination is, and *K* is no more than the total types of data. It is more general and extensible defining data types as *K*. Let k(k=1,2,⋯,K) express the priorities of different kinds of data and the data k=1 have the highest priority. Data k=K have the lowest priority. Let $\beta_{ki}\left(i=0,1,\cdots ,m\right)$ express the probability that the backoff stage *i* resets to 0 when the data (type *k*) are transmitted successfully, $\beta_{1i}>\beta_{2i}>\cdots >\beta_{Ki}$. Each type of data follows the independent backoff process, which means when a node wants to send a certain type of data, it only depends on the previous backoff process of that type of data and the current network status. EDCA defines different values of $CW_{\min}$ and $CW_{\max}$ for different access categories. The access category with smaller value of $CW_{\min}$ and $CW_{\max}$ has higher priority. While in this paper, the backoff schemes of all data have the same value of $CW_{\min}$ and $CW_{\max}$. To distinguish the priorities, when the data are transmitted successfully, the backoff stage of data with higher priority will reset to 0 with higher probability and the backoff stage of data with lower priority will remain unchanged with higher probability. In this way, the data priorities can be distinguished without defining different $CW_{\min}$ and $CW_{\max}$, which may cause worse system performance when CW cannot satisfy with the number of the vehicles. When the data fail to transmit, the backoff stage increases and the contention window size increases according to the BEB or QB alogrithm. When the data are transmitted successfully, the backoff stage resets to 0 with the calculated probability $\beta_{ki}$. The process is as follows:The vehicle node identifies the type of data *k*;The backoff process of the current data starts with the backoff series of the previous backoff process of similar data. Assuming that after similar data are transmitted successfully, the backoff stage sets to *i*, then the initialized contention window of the new data is $CW_{i}=rand\left(0,W_{i}-1\right)$;The node monitors the channel and does the backoff. When the backoff window decreases to 0, the node sends the data.If a channel collision occurs at the backoff stage i<m, the backoff stage adds 1. If a channel collision occurs at the backoff stage i=m, the backoff stage remains *m*. The node chooses the backoff window again and does the backoff;If the data are transmitted successfully, the vehicle node calculates $\beta_{ki}$ according to the previous success rate and resets the backoff stage to 0 with a probability of $\beta_{ki}$. (Calculating the $\beta_{ki}$ is introduced in the next section).

In the algorithm, different types of data do different backoff processes. When a channel collision happens, the contention window multiplicative increases. When data are transmitted successfully, the data with higher priority will reset their backoff stage to 0 with a higher probability. When the data with lower priority ensure their own transmission requirements, they make concessions to the data with higher priority, and the access contention will decrease so the data with higher priority can access the channel with a lower delay.

## 4. Modeling and Performance Analysis

### 4.1. State Transition Probability

Two-dimensional Markov Chain is used in this section to analyze the proposed algorithm. Let (i,j) express the backoff state of the node. In the formula, *i* is the backoff stage, which is the double times of the node’s contention window, $W_{i}=2^{i}W_{0}\left(0\leq i\leq m\right)$ in BEB or $W_{i}={\left(1+i\right)}^{2}W_{0}\left(0\leq i\leq m\right)$ in QB; *j* is the value of the backoff window at the current time, $0\leq j\leq W_{i}-1$. It can be found that the node’s backoff state space is $\Omega =\left\{\left(i,j\right)|0\leq i\leq m,0\leq j\leq W_{i}-1\right\}$. Let $\left(s(t),b(t)\right)$ express the backoff state of the node at the moment *t*, define the transition probability as
(1)$$P\left\{i^{\prime },j^{\prime }|i,j\right\}=P\left\{s\left(t+1\right)=i^{\prime },b\left(t+1\right)=j^{\prime }|s\left(t\right)=i,b\left(t\right)=j\right\}$$

Then the two-dimensional stochastic process $\left(s(t),b(t)\right)$ is a discrete two-dimensional Markov Chain with state space Ω. To simplify, let $b_{i,j}=\lim_{t\to \infty }\;P\left\{s(t)=i,b(t)=j\right\}$ express the steady state probability of the Markov Chain.

Assuming the number of vehicles in one-hop range follows a Poison Point Process with parameter λ, so the estimated number of contention nodes follows the distribution that $P\left\{n=k\right\}=\frac{\lambda^{k}e^{-\lambda }}{k!}\left(k=0,1,2\cdots \right)$. Denote the probability that the nodes have data to send is τ and the collision probability is *p*. Obviously, the collision probability is equal to the probability that at least one of the n−1 nodes transmit data at the same moment, $p=1-{\left(1-\tau \right)}^{n-1}$, so $p=\sum_{k=1}^{\infty }\frac{\lambda^{k}e^{-\lambda }}{k!}\left[1-{\left(1-\tau \right)}^{k-1}\right]=1-e^{-\lambda }-\frac{e^{-\lambda \tau }}{1-\tau }+\lambda e^{-\lambda }$.

The state transition probability of the Markov Chain is shown in Figure 2. The size of the contention window doubles each time when the channel collision happens until it reaches the maximum backoff stage *m*. When the data transmitted successfully, the backoff stage will reset to 0 with a calculated probability. To simplify the derivation, in this paper, let $\beta_{k}=\beta_{ki}\left(i=0,1,\cdots ,m\right)$.

It can be seen in the figure,
(2)$$\left\{\begin{array}{c} P\left\{i,j|i,j+1\right\}=1\;\;i\in \left[0,m\right],j\in \left[0,W_{0}-2\right]\\ P\left\{0,j|0,0\right\}=\frac{1-p}{W_{0}}\;\;i=0,j\in \left[0,W_{0}-1\right]\\ P\left\{i,j|i,0\right\}=\frac{\left(1-p\right)\left(1-\beta_{k}\right)}{W_{i}}\;\;i\in \left[1,m-1\right],j\in \left[0,W_{i}-1\right]\\ P\left\{m,j|m,0\right\}=\frac{\left(1-p\right)\left(1-\beta_{k}\right)+p}{W_{m}}p\;\;i=m,j\in \left[0,W_{m}-1\right]\\ P\left\{i,j|i-1,0\right\}=\frac{p}{W_{i}}\;\;i\in \left[1,m\right],j\in \left[0,W_{i}-1\right]\\ P\left\{0,j|i,0\right\}=\frac{\left(1-p\right)\beta_{k}}{W_{0}}\;\;i\in \left[1,m\right],j\in \left[0,W_{0}-1\right] \end{array}\right.$$

In the formula, the first item expresses the backoff window minus 1 each time; the second item expresses the node choosing the contention window again when it transmits data successfully at backoff stage 0; the third item expresses the node choose contention window again at stage *i* when it transmit data successfully; the fourth item expresses the node choosing the contention window again at stage *m*; the fifth item expresses the node doubling the size of the contention window when the channel collision happens and the sixth item expresses the node reseting the backoff stage to 0 when it transmits data successfully.

It also can be found in the Figure 2 that,
(3)$$\left\{\begin{array}{c} b_{0,0}^{\left(k\right)}\cdot p=\sum_{i=1}^{m}b_{i,0}^{\left(k\right)}\cdot \left(1-p\right)\beta_{k}\\ b_{i,0}^{\left(k\right)}\cdot \left[p+\left(1-p\right)\beta_{k}\right]=b_{i-1,0}^{\left(k\right)}\cdot p\;\;i\in \left[1,m-1\right]\\ b_{m,0}^{\left(k\right)}\cdot \left(1-p\right)\beta_{k}=b_{m-1,0}^{\left(k\right)}\cdot p \end{array}\right.$$

It can be obtained from the above formula that
(4)$$b_{i,0}^{\left(k\right)}=\frac{p}{p+\left(1-p\right)\beta_{k}}b_{i-1,0}^{\left(k\right)}={\left[\frac{p}{p+\left(1-p\right)\beta_{k}}\right]}^{i}b_{0,0}^{\left(k\right)}$$
(5)$$b_{m,0}^{\left(k\right)}=\frac{p}{\left(1-p\right)\beta_{k}}b_{m-1,0}^{\left(k\right)}=\frac{p}{\left(1-p\right)\beta_{k}}\cdot {\left[\frac{p}{p+\left(1-p\right)\beta_{k}}\right]}^{m-1}b_{0,0}^{\left(k\right)}$$

Let $H_{p}=\frac{p}{p+\left(1-p\right)\beta_{k}}$, then (4) and (5) can be simplified as
(6)$$b_{i,0}^{\left(k\right)}=H_{p}^{i}\cdot b_{0,0}^{\left(k\right)},\;\;i\in \left[1,m-1\right]$$
(7)$$b_{m,0}^{\left(k\right)}=\frac{p\cdot H_{p}^{m-1}}{\left(1-p\right)\beta_{k}}b_{0,0}^{\left(k\right)}$$

From Figure 2, we also can find that,
(8)$$b_{i,j}^{\left(k\right)}=\frac{W_{i}-j}{W_{i}}\cdot \left\{\begin{array}{c} \sum_{a=1}^{m}b_{a,0}^{\left(k\right)}\cdot \left(1-p\right)\beta_{k}+b_{0,0}^{\left(k\right)}\cdot \left(1-p\right)\;\;i=0\\ b_{i-1,0}^{\left(k\right)}\cdot p+b_{i,0}^{\left(k\right)}\cdot \left(1-p\right)\left(1-\beta_{k}\right)\;\;i\in \left[1,m-1\right]\\ b_{m-1,0}^{\left(k\right)}\cdot p+b_{m,0}^{\left(k\right)}\cdot \left[\left(1-p\right)\left(1-\beta_{k}\right)+p\right]\;\;i=m \end{array}\right.$$

According to Formulas (3) and (8), it can be obtained that
(9)$$b_{i,j}^{\left(k\right)}=\frac{W_{i}-j}{W_{i}}\cdot b_{i,0}^{\left(k\right)},\;\;i\in \left[0,m\right],j\in \left[0,W_{i}-1\right]$$

As $b_{i,j}^{\left(k\right)}$ is the steady state probability of the Markov Chain, the sum of the probability must be 1, so
(10)$$1=\sum_{i=0}^{m}\sum_{j=0}^{W_{i}-1}b_{i,j}^{\left(k\right)}=\sum_{i=0}^{m}b_{i,0}^{{}^{\left(k\right)}}\sum_{j=0}^{W_{i}-1}\frac{W_{i}-j}{W_{i}}=\frac{1}{2}\sum_{i=0}^{m}b_{i,0}^{{}^{\left(k\right)}}\left(1+W_{i}\right)$$

Plug (6), (7) into (10), then
(11)$$b_{0,0}^{\left(k\right)}=\frac{2}{\sum_{i=0}^{m-1}H_{p}^{i}\left(1+W_{i}\right)+\frac{p\cdot H_{p}^{m-1}}{\left(1-p\right)\beta_{k}}\left(1+W_{m}\right)}$$

The probability that the node transmits the data *k* is
(12)$$\tau \left(k\right)=\sum_{i=0}^{m}b_{i,0}^{{}^{\left(k\right)}}=\frac{p+\left(1-p\right)\beta_{k}}{\left(1-p\right)\beta_{k}}b_{0,0}^{{}^{\left(k\right)}}=\frac{b_{0,0}^{\left(k\right)}}{1-H_{p}}$$

Assume that in the system, the proportion of the emergency data is $\alpha_{0}$, and the proportion of the data *k* is $\alpha_{k}$, then the average transmission probability of the system is
(13)$$\tau =\sum_{k=1}^{K}\alpha_{k}\sum_{i=0}^{m}b_{i,0}^{\left(k\right)}=\sum_{k=1}^{K}\frac{\alpha_{k}\cdot b_{0,0}^{\left(k\right)}}{1-H_{p}}$$

### 4.2. Throughput and Delay

Denote *S* as the normalized network performance, which is equal to dividing the payload of the data transmitted in one frame by the average slot length [6].
(14)$$S=\frac{P_{s}P_{tr}E\left(P\right)}{\left(1-P_{tr}\right)\sigma +P_{s}P_{tr}T_{s}+\left(1-P_{s}\right)P_{tr}T_{c}}$$

In the formula, E(P) is the expectation of the payload length of the data. $T_{s}$ and $T_{c}$ are the average channel occupied time when the data transmitted successfully or failed respectively. $P_{tr}$ is the probability that there is at least one node sending data at any time, $P_{tr}=\sum_{k=1}^{\infty }\frac{\lambda^{k}e^{-\lambda }}{k!}[1-{\left(1-\tau \right)}^{k}]=1-e^{-\lambda }-e^{-\lambda \tau }+\lambda e^{-\lambda }\left(1-\tau \right)$. $P_{s}$ is the probability that there is only one node sends data, $P_{s}=\frac{\sum_{k=1}^{\infty }\frac{\lambda^{k}e^{-\lambda }}{k!}k\tau {\left(1-\tau \right)}^{k-1}}{P_{tr}}=\frac{\lambda \tau e^{-\lambda \tau }}{P_{tr}}$. The diagram of $T_{s}$ and $T_{c}$ are shown in Figure 3. As in the vehicular adhoc network, if the distance between two vehicles is more than one-hop range and less than two-hop range, when they both transmit data to the same node in their one-hop neighbor list, a hidden terminal problem happens. Using RTS/CTS protocol, these two nodes can monitor the collision when they cannot receive the CTS and will not waste time to transmit data.

The average delay $Del_{k}$ is the expectation time that the data waits to be sent until the transmission succeeds, it can be expressed as,
(15)$$Del_{k}=E\left(k\right)E_{s}$$

The $E_{s}$ is the average length of the time slot; it is the denominator of Formula (14). E(k) is the average slots that the data *k* need to be transmitted successfully,
(16)$$E\left(k\right)=\sum_{i=0}^{m}D_{i}q_{k,i}$$

In Formula (15), $q_{i}$ is the probability that the data transmitted at the backoff stage *i*, $D_{i}$ is the expectation time that the data waits to send at the backoff stage *i*,
(17)$$D_{i}=\sum_{j=0}^{i}\frac{W_{j}+1}{2}$$

Let $A_{k,i}$ express the probability that when new data arrive, they access the stage *i* and choose contention window, so
(18)$$A_{k,i}=\left\{\begin{array}{c} b_{0,0}\cdot \frac{W_{0}+1}{2}+\sum_{a=1}^{m}b_{a,0}\cdot \frac{W_{a}+1}{2}\cdot \beta_{k}\;\;i=0\\ b_{i,0}\cdot \frac{W_{i}+1}{2}\cdot \left(1-\beta_{k}\right)\;\;i\in \left[1,m\right] \end{array}\right.\;$$

And the probability that the data arrive backoff stage *i* and transmitted successfully is
(19)$$q_{k,i}=\left\{\begin{array}{c} {\sum_{a=0}^{i}A}_{k,a}\cdot p^{i-a}\;\;i\in \left[0,m-1\right]\\ \sum_{a=0}^{m}A_{k,a}\cdot \frac{p^{m-a}}{\left(1-p\right)\beta_{k}}\;\;i=m \end{array}\right.$$

Plug (16), (17), (19) into (15), we can obtain the expression of delay $Del_{k}=E_{s}\cdot \sum_{i=0}^{m}\sum_{j=0}^{i}\frac{W_{j}+1}{2}\cdot q_{k,i}$.

### 4.3. Delay Optimization

As said in the above section, the delay expression includes *n*, τ, *p*, $W_{0}$, $W_{i}$, $\beta_{k}$ and other parameters. *n* is the number of vehicles, it cannot be controlled. $W_{0}$, $W_{i}$ are the system parameters, usually not changed. τ, *p* are related to $\beta_{k}$, so the delay optimization is to optimize the value of $\beta_{k}$.

By optimizing $\beta_{k}$, we want to obtain the lowest average total delay with the delay limitation of each data set. So the optimization problem is:(20)$$\begin{array}{c} \min\;\sum_{k=1}^{K}\alpha_{k}E\left(k\right)\cdot E_{s}\\ s.t.\;Del_{k}\leq \theta_{k}\cdot D_{\min}(k)\;k\in \left[1,K\right]\\ \;0\leq \beta_{k}\leq 1\;k\in \left[1,K\right] \end{array}$$

In the formula, $\sum_{k=1}^{K}\alpha_{k}E\left(k\right)\cdot E_{s}$ is the average total delay, $D_{\min}(k)$ is the lowest delay of data *k* in the certain scene with given number of vehicles. So $Del_{k}\leq \theta_{k}\cdot D_{\min}(k)$ expresses the delay limitation of data *k*. $\theta_{k}$ distinguish the priority of data, $\theta_{1}<\theta_{2}<\cdots <\theta_{K}$.

To solve this optimization problem, $D_{\min}(k)$ should be obtained firstly. We can get $D_{\min}(k)$ by establishing the other optimization problem:(21)$$\begin{array}{c} \min\;E\left(k\right)\cdot E_{s}\;k\in \left[1,K\right]\\ s.t.\;0\leq \beta_{k}\leq 1\;k\in \left[1,K\right] \end{array}$$

It is easily found that $D_{\min}(k)$ can be obtained when $\beta_{k}=1$ and other β=0. Substituting to (16)–(19), the $D_{\min}(k)$ is derived.

Formula (20) is a nonlinear multivariate optimization problem, the expression is complex and nests massive fractions and high order square data. It is difficult to derive the explicit solution. Particle swarm optimization (PSO) starts with a random solution and searches for the optimal solution by iteration. The fitness is used to evaluate the quality of the solution, and the global optimum is searched by following the optimal value of the current search. This algorithm is easy to implement with high precision and fast convergence. So this paper uses PSO to solve this nonlinear multivariate optimization problem as follows,
$$\begin{array}{c} \min\;f\left(\beta_{k}^{\left(x\right)}\right)=E_{s}\cdot \sum_{i=0}^{m}\sum_{j=0}^{i}\frac{W_{j}+1}{2}\cdot q_{k,i}\left(\beta_{k}^{\left(x\right)}\right)\\ s.t.\;Del_{k}\leq \theta_{k}\cdot D_{\min}(k)\;k\in \left[1,K\right]\\ \;0\leq \beta_{k}\leq 1\;k\in \left[1,K\right] \end{array}$$

In the PSO, each individual is considered as a particle, and each particle represents a potential solution. The algorithm uses individual sharing of information, making the search of the whole group from disorder to order, so as to get the optimal solution. In this problem, there are *K* parameters to be solved $\beta_{1}\;\;\;∼\;\;\;\;\beta_{K}$, so the particle swarm consists with *K* particles. Each particle has its own position and speed, which denote the searching distance and direction. Define $\beta_{k}^{\left(x\right)}$ is the searching solution after the data *k* searching *x* times. $V_{k}^{\left(x\right)}$ is the searching speed when the data *k* search *x* times. $Best_{k}^{\left(x\right)}$ is the current best solution after the data *k* searching *x* times. Then the algorithm can be described with $\left(\beta_{k}^{\left(x\right)},V_{k}^{\left(x\right)},Best_{k}^{\left(x\right)}\right)$:The searching solution at current time$\beta_{k}^{\left(x\right)}\;=\;(\beta_{1}^{\left(x\right)},\beta_{2}^{\left(x\right)},\beta_{3}^{\left(x\right)},\cdots ,\beta_{K}^{\left(x\right)})$;The searching speed at current time$V_{k}^{\left(x\right)}\;=\;\left(V_{1}^{\left(x\right)},V_{2}^{\left(x\right)},V_{3}^{\left(x\right)},\cdots ,V_{K}^{\left(x\right)}\right)$;The best solution at current time$Best_{k}^{\left(x\right)}\;=\;(B_{1}^{\left(x\right)},B_{2}^{\left(x\right)},B_{3}^{\left(x\right)},\cdots ,B_{K}^{\left(x\right)})$;

The algorithm process is as follows in Table 1.

There are some variable parameters that need to be set in the algorithm. $c_{1}$ is learning factor, appropriate adjustments can minimize the local minima value and speed up the convergence, the usual range is [0,4], in this paper $c_{1}=2$. $V_{\max}$ is the variable maximum searching speed, it is used to control the convergence speed, when $|V_{k}^{\left(x\right)}\left|>\right|V_{\max}|$, the searching speed choose $|V_{\max}|$, in this paper $|V_{\max}|=0.01$. The bigger weight value *w* is, the easier it will be to inherit the current search speed and the stronger global search ability. With a small weight value *w*, the algorithm tends to local search and converges fast, but may terminate in local optima value. Therefore, the appropriate method is to set larger weight value at the initial stage of the iteration, in order to improve the global search capability and to reduce the weight value at the later stage to speed up the convergence. In this paper, $w=w_{\max}\cdot \theta^{x-1}$, in the formula, $w_{\max}=1$, θ=0.5.

## 5. Simulation Results and Analysis

In this section, we simulate the proposed algorithm with Matlab, and compared with QB algorithm [11] and FMC-MAC [14]. The parameters of the simulation are shown in Table 2 and Table 3 [6]. In addition, the network performance and the optimal value of $\beta_{k}$ are analyzed with the given parameters.

Figure 4 shows the average total delay with the different number of vehicles. By comparing Figure 4a–c, it can be seen intuitively that the average total delay increases with the increasing number of vehicles. In addition, we can find that when the number of vehicles is smaller (n<40), the lowest delay can be obtained at $\beta_{1}\;=\;\beta_{2}\;=\;1$, which are the same as the BEB and QB algorithm. While when the road is crowed (n>50 ), the original backoff models like the BEB or QB model cannot satisfy the request of mass data transmission, the performance becomes poor because the transmission collisions increase. By using the adaptive backoff scheme, we can adjust the backoff parameters of different kinds of data, like $\beta_{1}$ and $\beta_{2}$ in the Figure 4. In this way, the average total delay decreases with the less transmission collisions. The adaptive backoff scheme is helpful to both EBE and QB algorithm. In particular, without the adaptive backoff scheme, the QB algorithm is better than BEB algorithm. While after using the adaptive backoff scheme, the performance of the enhanced BEB algorithm is better than the enhanced QB algorithm. So this adaptive backoff scheme is more suitable for the BEB algorithm.

Figure 5 shows the optimal value of $\beta_{k}$ both in the enhanced BEB and enhanced QB. In order to make the image simple and clear, there are two kinds of data in the simulation, so two parameters $\beta_{1}$ and $\beta_{2}$ need to be solved. Assuming that the delay requirement of data 1 is $Del_{1}\leq \theta_{1}\cdot D_{\min}(1),\;\theta_{1}\;=\;1.15$, and the delay requirement of data 2 is $Del_{2}\leq \theta_{2}\cdot D_{\min}(2),\;\theta_{2}=1.3$. As the priority of data 1 is higher than data 2, the assumed parameters $\theta_{1}<\theta_{2}$. Both in the enhanced BEB and QB, we can find that $\beta_{1}$ is always larger than $\beta_{2}$, because the delay requirement of data 1 is higher than data 2. To meet this requirement, the data 1 needs higher probability to reset the backoff stage to 0 compared with the data 2, so $\beta_{1}>\beta_{2}$. In the enhanced BEB, the $\beta_{k}$ decreases when n>44. On the other hand, in the enhanced QB, the $\beta_{k}$ decreases when n>48, which conforms to Figure 4. For the same optimization target and system parameters, when the number of nodes in the network increases, the network load and transmission collision probability increase. In order to reduce the data transmission conflict and transmission delay, $\beta_{k}$ will reduce accordingly, so the node will choose the contention window at the higher backoff stage to reduce the transmission probability. In particular, although the data 2 make a concession as their priority are lower, $\beta_{2}$ has a lower limit because their delay has constraints.

Figure 6 shows the saturated throughput performance of the proposed algorithm compared with the IEEE 802.11p, QB algorithm and FMC-MAC. The system throughput decreases along with the increasing number of nodes. This is because when the system parameters are given, the probability of the transmission collision increases with the increasing number of nodes, which reduces the system throughput. It can be seen in the figure that the saturated throughput of IEEE 802.11p algorithm does not decrease significantly. The reason is that the saturated throughput is the maximum throughput that can be reached when the number of nodes is given, without considering the limitation of maximum backoff stage and initial contention window on the data transmission probability. So if the system gives the number of nodes, maximum backoff stage *m* and initial contention window $W_{0}$, the saturated throughput is not always reachable. Because of lack of flexibility, in some special networks, the actual throughput will be much less than the saturated throughput. The saturated throughput of the FMC-MAC [14] protocol is higher than the standard IEEE 802.11p. Because they allocate more resources for non-safety applications when the resources are enough for allocation, they fail to guarantee the reliability of safety applications. After using the adaptive backoff algorithm, the actual throughput is less than the theoretical achievable saturated throughput, but the losing performance is not much, and the actual throughput is larger than 90% saturated throughput.

The comparison of average total delays in two different optimization models is shown in Figure 7. The reference line of the contrast is the delay of IEEE 802.11p, QB algorithm and FMC-MAC. Without the proposed adaptive backoff scheme, the performance of IEEE 802.11p is worse than FMC-MAC and QB algorithm, and FMC-MAC will have a better performance compared with QB in the high density network. After using the adaptive backoff scheme, the delay of IEEE 802.11p and QB can decrease and become better than FMC-MAC. We can find that by using the adaptive backoff scheme in the IEEE 802.11p and QB, the average total delay can be optimized when the road is crowed, especially in the eBEB algorithm (the delay can reduce nearly 15%). Compare IEEE 802.11p with QB, the performance of QB is always better than IEEE 802.11p. While after using the adaptive backoff scheme, the eBEB algorithm is better than the eQB algorithm when n>52. So the adaptive backoff scheme is more applicable in BEB than in QB.

In Figure 8a,b, the optimal delay is compared with the maximum and minimum delay in the same scene. In the assuming scene, the priority of data 1 is higher than data 2 and the ratio of data 1 is lower than data 2. According to the disparity of the maximum and minimum delay, we can find that the disparity of data with larger ratio is smaller than the data with smaller ratio. It means that the delay of the data with larger ratio is difficult to optimize. In both the BEB and QB algorithm, when the number of vehicles is smaller than a certain threshold, there is no need to use the adaptive backoff scheme. When the number of vehicles becomes larger (In BEB, n>44 or in QB, n>48), the proposed adaptive backoff scheme can modify the data delay according to their upper limit. Overall, considering all kinds of data delay, the adaptive backoff scheme can reduce the average total delay.

## 6. Conclusions

In this paper, an adaptive backoff algorithm is proposed to distinguish traffic priority, especially for the autonomous vehicular network. The algorithm classifies the data according to their different delay requirements and designs different backoff processes for them. The backoff parameters are adjusted to optimize the delay according to the network performance. This algorithm has the characteristics of real time, controllability and insists on multi class service. The simulation results show that the average total delay is reduced when the road is crowded, and each kind of data delay is optimized without losing much system throughput. The proposed algorithm is an adaptive backoff algorithm for vehicular networks.

## Figures and Tables

**Figure 1 sensors-18-04421-f001:**
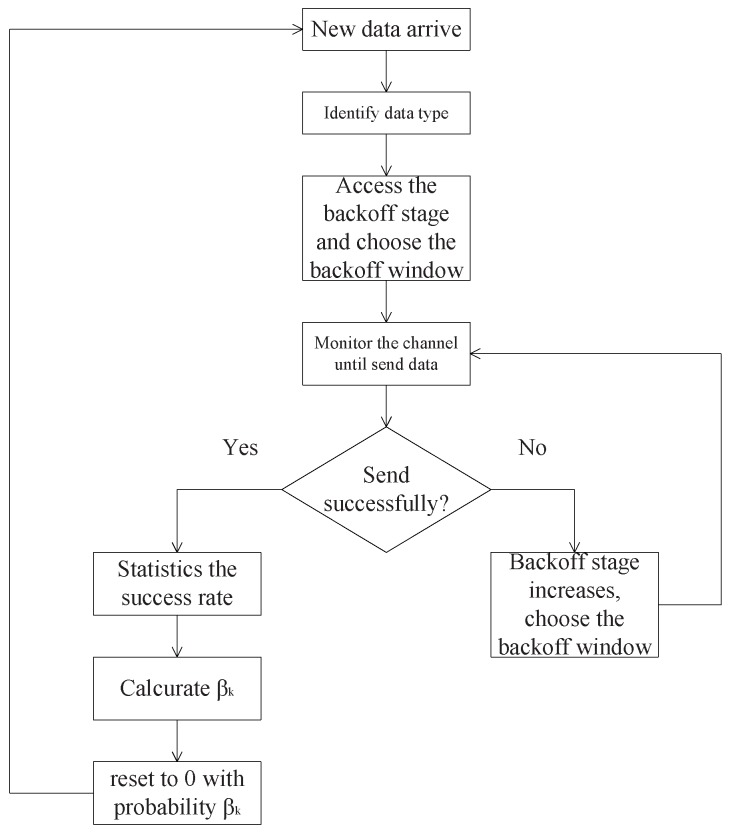
The transmission process using adaptive backoff scheme.

**Figure 2 sensors-18-04421-f002:**
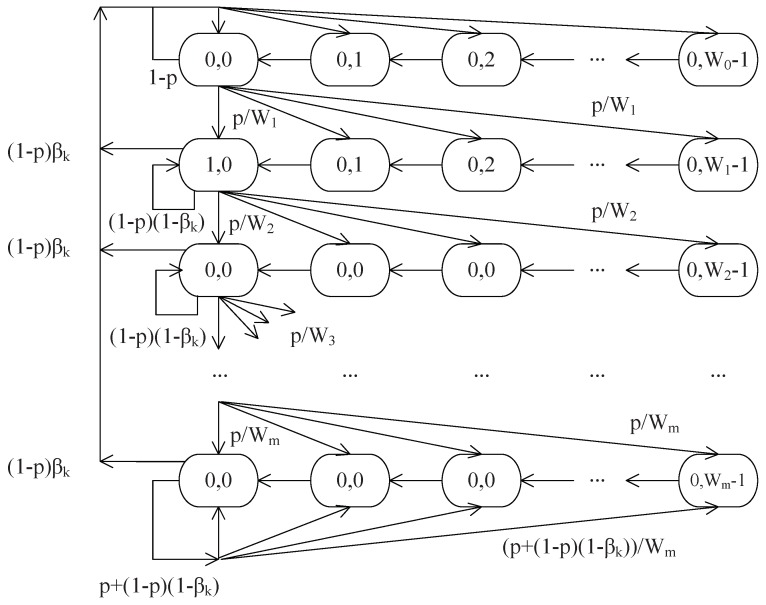
The state transition probability of the Markov Chain.

**Figure 3 sensors-18-04421-f003:**
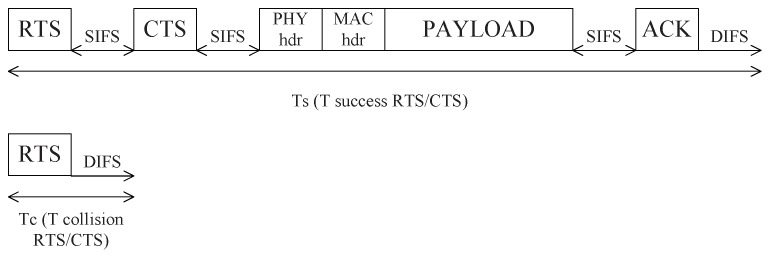
The diagram of $T_{s}$ and $T_{c}$.

**Figure 4 sensors-18-04421-f004:**
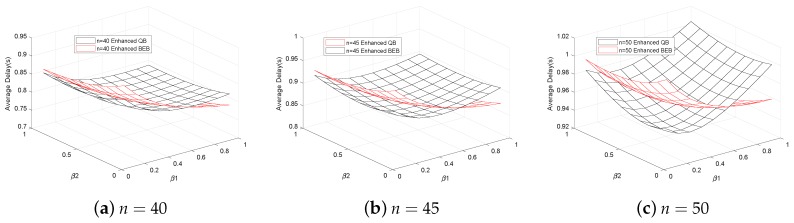
Average total delay with different number of vehicles.

**Figure 5 sensors-18-04421-f005:**
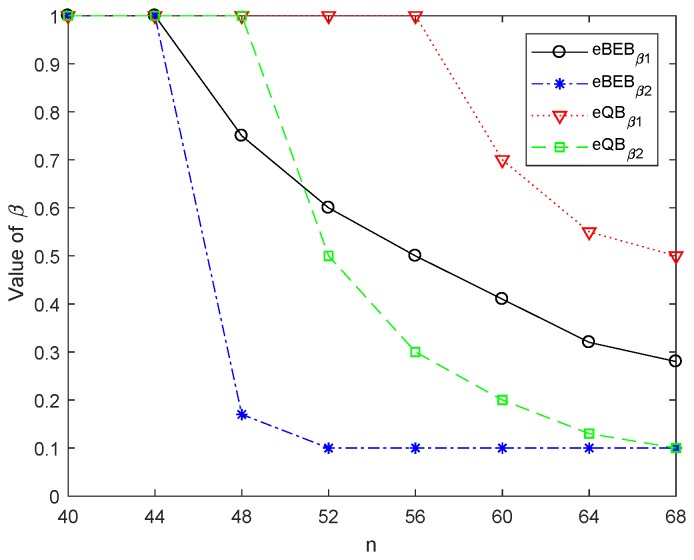
Optimal value of $\beta_{k}$ of PSO optimization process.

**Figure 6 sensors-18-04421-f006:**
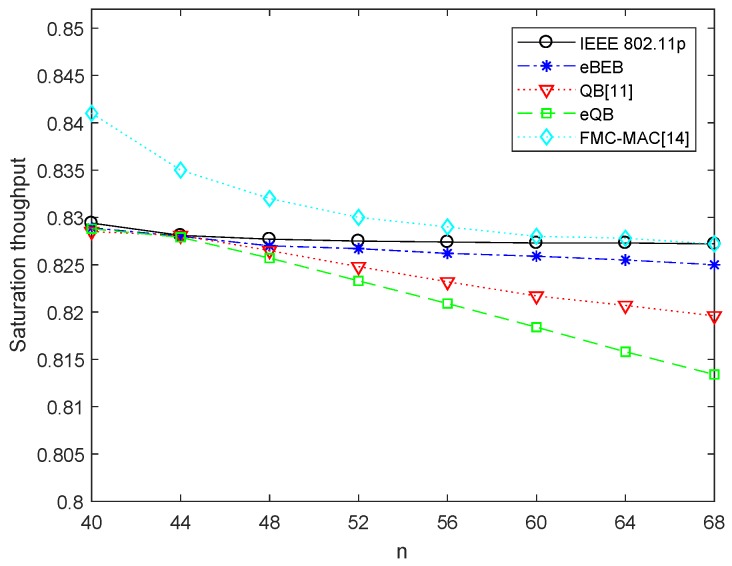
The saturated throughput.

**Figure 7 sensors-18-04421-f007:**
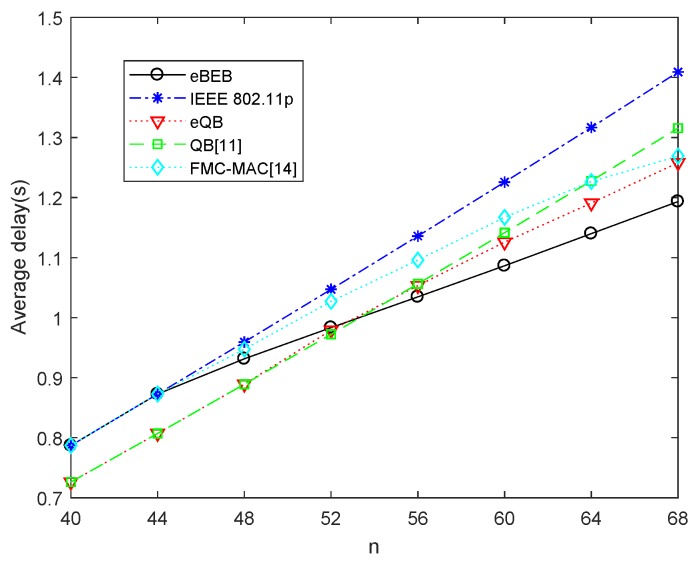
The optimal value of the average total delay.

**Figure 8 sensors-18-04421-f008:**
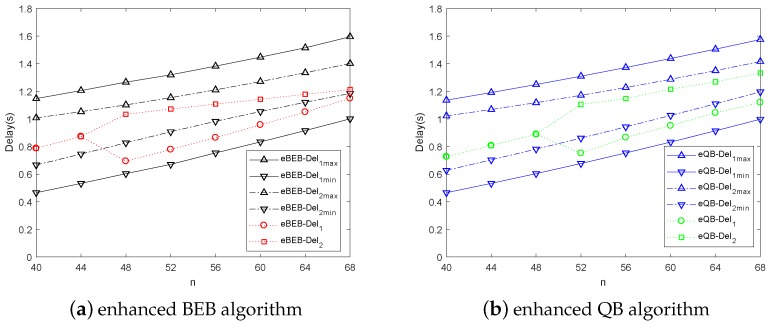
The maximum, minimum and optimal delay of the adaptive backoff scheme.

**Table 1 sensors-18-04421-t001:** Particle swarm optimization (PSO) Algorithm.

Algorithm Process
1. Initialize the search step $V_{k}^{\left(x\right)}\in \left[-V_{\max},V_{\max}\right]$, parameters $\beta_{k}^{\left(x\right)}\in \left[0,1\right]$, weight value *w* and accuracy requirement ε;
2. Let $\beta_{1}=1$, $\beta_{k}=0\left(k=2\tilde{\;}K\right)$, calculate $D_{\min}(1)$. In the same way, calculate other $D_{\min}(k)$;
3. Set $\beta_{k}^{\left(0\right)}=0$, perform the first search, x=1, $V_{k}^{\left(1\right)}=0.001$, $\beta_{k}^{\left(1\right)}=0.001$, $Best_{k}^{\left(1\right)}=\beta_{k}^{\left(1\right)}$;
4. Do while $|f\left(\beta_{k}^{\left(x\right)}\right)-f\left(\beta_{k}^{\left(x-1\right)}\right)|>\varepsilon$;
5. x=x+1;
6. Compute the initial optimal solution $Best_{k}^{\left(x\right)}$, $Best_{k}^{\left(x\right)}=\left\{\begin{array}{c} \beta_{k}^{\left(x\right)}\;\mathrm{if}\;f\left(\beta_{k}^{\left(x\right)}\right)<f\left(\beta_{k}^{\left(x-1\right)}\right)\\ Best_{k}^{\left(x-1\right)} \end{array}\right.$;
7. Computer the search step, $V_{k}^{\left(x+1\right)}=w\cdot V_{k}^{\left(x\right)}+c_{1}\cdot rand\left(1\right)\cdot \left(Best_{k}^{\left(x\right)}-\beta_{k}^{\left(x\right)}\right)$;
8. Compute the parameter $\beta_{k}^{\left(x+1\right)}=\beta_{k}^{\left(x\right)}+V_{k}^{\left(x+1\right)}$;
9. If the constraint conditions $Del_{k}\leq \theta_{k}\cdot D_{\min}(k)$ cannot be satisfied, go back to step 7, or else go on;
10. End do
11. Return $\beta_{k}^{\left(x\right)}$ and $f(\beta_{k}^{\left(x\right)})$

**Table 2 sensors-18-04421-t002:** Moving parameters of highway scene.

Parameter	City
Number of lanes	4
Length of road	1 km
number of directions	2
Width of lanes	5 m
Mean speed	40 km/h
Variance of speed	10 km/h
GPS time interval	0.1 s
Communication range	200 m
Number of vehicles (*N*)	0∼68

**Table 3 sensors-18-04421-t003:** System channel Parameters.

Parameter	Value
DIFS (μs)	128
SIFS (μs)	28
MAC header (bits)	272
PHY header (bits)	128
ACK	112 + PHY header
Propagation delay (μs)	1
Timeslot length σ (μs)	20
packet size *E(P)* (bytes)	1024
Channel data rate (Mbit·s${}^{-1}$)	1
$CW_{\min}$	6∼96

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
