# Peer review of "A QoS Based Adaptive Backoff Scheme for Vehicular Ad Hoc Networks"

_sensors, 2018, doi:10.3390/s18124421_

Reviewer 1 Report

This paper presents a priority-based adaptive backoff scheme for vehicular ad hoc networks. The proposed scheme is interesting. Authors also analyze the proposed scheme.

Followings are some comments and suggestions to improve paper:

For vehicular networks, data priority has been specified in DSRC/WAVE standard set, e.g., IEEE P1609.3 defines 8 priority levels. This paper considers k data types. Definition of each data type should be useful.

In DSRC/WAVE and 802.11e, based on priority level, backoff parameters are also different. What is key difference between the proposed backoff and existing backoff?

Section 3 states that "If a channel collision occurs at the backoff stage i < m, the backoff stage plus 1". Using ACK mechanism, a node only knows that transmission is successful or not. Non-successful transmission is not necessarily a collision, e.g., destination vehicle moves away.

Section 4.1 assumes a node has n neighbors, however, the derived collision probability p is independent of n. Describe relationship between and p and n.   

Figure 3 uses RTS/CTS, describe the suitability of using RTS/CTS in vehicular communication. 

Figure 4 only shows n = 40, 45 and 50, how were conclusions for n < 40 and n > 50 reached?

In section 4.1, "define the transition probability is" -> "define the transition probability as"

Section 5, "it is can be seen" -> "it can be seen"

Author Response

Response to Reviewer #1:

We are very grateful to your comments for the manuscript, and each of your suggestions has great importance for our research work and thesis writing. According to your advice, we amended the relevant part in manuscript. We also have checked and revised the manuscript in accordance with your comments, and carefully proof-read the whole manuscript to minimize typographical, grammatical, and bibliographical errors. Each of your questions was answered below.

1. Thank you for your kind recommendation. We add the related description in section 3 as follows:

In IEEE 802.11e, EDCA algorithm defines eight kinds of traffic categories (TC) and four kinds of access categories (AC), which is also cited in DSRC/WAVE standard and IEEE 1609.3. Eight kinds of TC are mapped to the queues of four kinds of AC, respectively. Similarly to the definition of EDCA, in the assuming vehicular adhoc network, the data are divided into K types according to their delay requirements. For example, if the data are just divided to high priority and low priority, in this case, the K equals to two. If we need four ACs like EDCA, the K equals to four etc. The larger K is, the higher the data discrimination is, and K is no more than the total types of data. It is more general and extensible defining data types as K.

2. We are sorry that we did not express the differences between DSRC/WAVE, 802.11e and the scheme in this paper clearly. You are right that it is inconvenient for readers to understand. We add the related description in section 3 as follows:

EDCA defines different values of  and  for different access categories. The access category with smaller value of  and  has higher priority. While in this paper, the backoff schemes of all data have the same value of  and . To distinguish the priorities, when the data are transmitted successfully, the backoff stage of data with higher priority will reset to 0 with higher probability and the backoff stage of data with lower priority will remain unchanged with higher probability. In this way, the data priorities can be distinguished without defining different  and , which may cause worse system performance when CW cannot satisfy with the number of the vehicles.

3. Thank you for your attention. We add the related description in section 2.1 as follows:

As the data transmission time is too little for vehicles to move very long distance, we can assume that all the vehicles can be seen still when they transmit or receive data. For example the speed of vehicle is 100km/h, the data transmission time is  level, the vehicle will move far less than 1 meter. When a vehicle begins transiting data, it judges the distance and chooses a nearer relay node or not according to its one-hop neighbor list. So the transmission collision happens only if the channel is used by other nodes at the same time.

4. We are sorry that we did not express the relationship between p and n clearly. We add the related description in section 4.1 as follows:

Assuming the number of vehicles in one-hop range follows a Poison Point Process with parameter , so the estimated number of contention nodes follows the distribution that . And collision probability .

5. Thank you for your kind recommendation. We add the related description in section 4.2 as follows:

In the vehicular adhoc network, if the distance between two vehicles is more than one-hop range and less than two-hop range, when they both transmit data to the same node in their one-hop neighbor list, a hidden terminal problem happens. Using RTS/CTS protocol, these two nodes can monitor the collision when they cannot receive the CTS and will not waste time to transmit data.

6. Thank you for your question. Compared with figure 7 and 8, it can be seen that when the number of vehicles is less than 40, the optimized value is equal to the original value because the given CW is enough when the number of vehicles is fewer. And we show this three pictures to express that when , the total delay is a convex optimization problem and it exists the optimized value.

7. Thank you for your advices. We have revised "define the transition probability is" to "define the transition probability as".

8. Thank you for your advices. We have revised "it is can be seen" to "it can be seen".

Thank you for precious time on our manuscript. Every of your comment is with great help to our paper. We hope to learn more knowledge from you. Thank you.

Reviewer 2 Report

2- The paper needs general language corrections. The quality of language must be improved. Some typos exist, some sentences need to be revised.

4- Why did you compare your work only with reference #11? Why didn’t you compare with other studies especially newer ones? Is only one work reported in the state of the art?

6- The idea in the paper is reasonable. I strongly recommend that the authors improve the manuscript significantly by addressing above mentioned issues.

Author Response

Response to Reviewer #2:

We are very grateful to your comments for the manuscript, and each of your suggestions has great importance for our research work and thesis writing. According to your advice, we amended the relevant part in manuscript. We also have checked and revised the manuscript in accordance with your comments, and carefully proof-read the whole manuscript to minimize typographical, grammatical, and bibliographical errors. Each of your questions was answered below.

1. Thank you for your kind recommendation. We have revised the manuscript according to the comments.

2. Thank you for your kind recommendation. We have checked and revised the manuscript in accordance with your comments, and carefully proof-read the whole manuscript to minimize typographical, grammatical, and bibliographical errors.

3. Thank you for your attention. We have adjusted the position of table 2,3 and figure 4.

4. Thank you for your question. In the original manuscript, we compared our work with IEEE 802.11p and QB[11] to prove that the presented backoff scheme can increase the performance both in IEEE 802.11p and QB. You are right that we should compare our work with more studies. In the figure 6 and 7 of the revised manuscript, we add a new performance curve of FMC-MAC[14] and add the related description as follows;

Figure 6 The saturation throughput

Figure 7 The optimal average total delay

Figure 6…, the saturated throughput of the FMC-MAC protocol is higher than the standard IEEE 802.11p. Because they allocate more resources for non-safety applications when the resources are enough for allocation, but they fail to guarantee the reliability of safety applications.…

The comparison of average total delays in two different optimization models is shown in figure 7. The reference line of the contrast is the delay of IEEE 802.11p, QB algorithm and FMC-MAC. Without the proposed adaptive backoff scheme, the performance of IEEE 802.11p is worse than FMC-MAC and QB algorithm, and QB algorithm will have a better performance compared with FMC-MAC in the high density network. After using the adaptive backoff scheme, the delay of IEEE 802.11p and QB can decrease and become better than FMC-MAC.…

5. Thank you for your kind recommendation. We have revised all the labels to make them more readable. Is

Thank you for precious time on our manuscript. Every of your comment is with great help to our paper. We hope to learn more knowledge from you. Thank you.

Round  2

Reviewer 2 Report

Congratulations.